# Exploring mtDNA Databases to Evaluate the Population Structure and Genetic Diversity of *Tursiops truncatus* in the Atlantic Ocean: Implications for the Conservation of a Small, Offshore Population

Brenda Godoy Alexandre [1,2], Marcelo Merten Cruz [3], Karina Bohrer do Amaral [4], Lilian Sander Hoffmann [5], Thales Renato Ochotorena de Freitas [1,2,*] and Rebeca Zanini [6,7,*]

1. Laboratório de Citogenética e Evolução Animal, Departamento de Genética, Universidade Federal do Rio Grande do Sul (UFRGS), Porto Alegre 91509-900, Brazil; brendagalex@gmail.com
2. Programa de Pós-Graduação em Genética e Biologia Molecular, Universidade Federal do Rio Grande do Sul, Porto Alegre 91509-900, Brazil
3. Centro Nacional de Pesquisa e Conservação da Biodiversidade Marinha do Sudeste e Sul do Brasil (CEPSUL), Instituto Chico Mendes de Conservação da Biodiversidade (ICMBio), Rio Grande 96207-480, Brazil; marcelomcruz4@gmail.com
4. Laboratório de Sistemática e Ecologia de Aves e Mamíferos Marinhos (LABSMAR), Universidade Federal do Rio Grande do Sul, Porto Alegre 91509-900, Brazil; karinabohrerdoamaral@gmail.com
5. Laboratório de Nectologia, Departamento de Oceanografia, Universidade Federal do Espírito Santo/UFES, Vitória 29075-900, Brazil; liliansander@gmail.com
6. Centre for Ecology, Evolution, and EnvironmentalChanges (cE3c), Faculdade de Ciências da Universidade de Lisboa, 1749-016 Lisbon, Portugal
7. iNOVA4Health, NOVA Medical School, Universidade Nova de Lisboa, 1169-056 Lisbon, Portugal
* Correspondence: thales.freitas@ufrgs.br (T.R.O.d.F.); rzanini@ciencias.ulisboa.pt (R.Z.)

**Abstract:** Inshore and offshore bottlenose dolphin, *Tursiops truncatus*, ecotypes were distinguished through genetics, distribution, diet, morphology, diversity, and social behaviors. Although *T. truncatus* is a widely studied species, few studies have focused on offshore populations. Offshore biodiversity is frequently neglected due to the difficulty of data collection, and therefore, it is challenging to assess how threatened these populations are. A small, offshore population of dolphins residing around the Saint Peter Saint Paul Archipelago (SPSPA) in the middle of the Atlantic Ocean has been monitored for several years, and a decrease in the number of dolphin sightings has recently been noticed. We analyzed a comprehensive mtDNA control-region sequence dataset for this species to infer the conservation status and better understand the relationships between the SPSPA population and other offshore populations. We assessed the genetic diversity and population structure of the bottlenose dolphin from inshore and offshore populations of the Atlantic Ocean. Offshore populations are more genetically diverse and have less variation between populations than inshore populations. The offshore populations share haplotypes, indicating potential gene flow. However, the SPSPA population presented the lowest levels of genetic diversity between populations. The conservation status of the SPSPA population is concerning, and it is necessary to apply effective management strategies to guarantee its protection.

**Keywords:** *Tursiops truncatus*; mtDNA control region; offshore populations; conservation genetics

## 1. Introduction

The study of the spatial distribution of genetic variability between and within populations is an important approach to molecular ecology and conservation genetics [1]. This knowledge is particularly beneficial for marine organisms, which are often challenging to study. Also, they are exposed to various anthropogenic impacts, such as habitat degradation and climate change [2], as is the case of cetaceans—the group of dolphins and whales.

Therefore, identifying and diagnosing how natural populations will respond to a changing world is crucial.

Dolphins are large top predators, playing an important role in maintaining the structure and function of their environment [3,4]. The bottlenose dolphin, *Tursiops truncatus* (Montagu, 1821), has a worldwide distribution in temperate and tropical waters [5,6]. Two ecotypes have been widely recognized across its distribution—offshore and inshore. These ecotypes can be differentiated on several levels, including ecology, morphology, and genetic traits. The distinction between the ecotypes for this species has been described in several geographic locations across the Atlantic and Pacific Oceans [7–11].

Studies in different regions have indicated that bottlenose dolphin inshore populations present low levels of genetic diversity and high population genetic structure on small geographic scales. On the other hand, offshore dolphins exhibit higher genetic diversity and lower potential for population genetic structure, even at considerably larger spatial scales [9–14]. Although *T. truncatus* is a widely studied species, many studies have primarily focused on inshore ecotypes, with a limited number of studies focused on offshore populations [12,15], mainly due to logistical constraints in offshore waters.

In the Northwestern Atlantic, a study integrating morphological and genetic analyses found evidence supporting species delimitation of *Tursiops erebennus*, restricted to coastal and estuarine waters. In contrast, the offshore group belongs to the worldwide species *T. truncatus* [16]. The two bottlenose dolphin ecotypes are distinct mitochondrial lineages, with lower genetic diversity in coastal populations [10]. A strong level of differentiation between coastal and pelagic dolphins was also found in the Northeast Atlantic region using microsatellite and mtDNA markers [10]. This finding was corroborated using microsatellites [17] and a genomic approach [14].

In the Caribbean and Gulf of Mexico, the distribution of the two ecotypes overlaps in several regions sampled [18]. The results suggest gene flow—the transfer of genetic material from one population to another—in the present or recent past, as some haplotypes described as belonging to the offshore ecosystem were shared between the Caribbean and the Azores [19].

In the South Atlantic Ocean, Costa et al. [20] found congruence between the morphological and genetic data, confirming the presence of two distinct ecotypes in the western South Atlantic—inshore and offshore ecotypes—with significant levels of evolutionary divergence. The offshore ecotype displayed greater genetic diversity in nuclear and mitochondrial DNA than the coastal ecotype, suggesting it as an offshore characteristic [9–11,13,14]. These results support the description of the coastal ecotype of this region as a subspecies: *Tursiops truncatus gephyreus* [11,20–23].

Although offshore populations are more genetically diverse, a genetic study based on microsatellite markers and mtDNA-CR analysis [15] observed that the diversity of haplotypes and nucleotides in a small, offshore population from the Saint Peter Saint Paul Archipelago (SPSPA) was lower than that found in the Azores and Madeira [12]. Individuals from the SPSPA share haplotypes with offshore individuals from the North Atlantic Ocean, suggesting a possible gene flow between these populations [15]. Microsatellite markers and mtDNA-CR also were employed to compare the offshore SPSPA population with inshore populations from the Brazilian coast [22], and the results indicate that oceanic SPSPA dolphins may be genetically isolated from inshore populations in the southwestern Atlantic Ocean.

The Saint Peter Saint Paul Archipelago is a small archipelago placed in offshore waters within the Brazilian Economic Exclusive Zone, located in the central equatorial Atlantic Ocean [24]. A population of bottlenose dolphins (approximately 30 individuals) resides around the SPSPA [25–27]. These dolphins have been under monitoring since 2005 through photo-identification, indicating that individuals from this group exhibit high site fidelity within the waters surrounding the SPSPA [26,27]. However, during the last monitoring, the animals were not seen occupying the area (Hoffmann et al., in prep), which raises concerns about the conservation status of this population.

Although there are many studies on *T. truncatus* populations in the Atlantic Ocean, most focus on more restricted areas or comparing few populations. Therefore, the dif-

ferences between ecotypes remain a subject of investigation. This knowledge gap holds significant implications for the effective conservation of these populations.

To better understand the population dynamics of *T. truncatus* in the Atlantic Ocean, we generated the most comprehensive mtDNA-CR dataset for this species by harnessing the information stored in databases. This marker was selected due to its extensive representation of several putative *Tursiops* populations in publicly available databases, such as NCBI. Rosel et al. [28]) state that mtDNA-CR has been the most chosen marker in cetacean genetic studies. It is the only marker with sufficient data available in many populations, subspecies, and species. For most studies, sample sizes were suitable, but adequate geographic sampling for broadly distributed taxa was often lacking.

We aim to compare diversity and pairwise differences among putative populations across Atlantic waters. Our goal is to enhance our understanding of the relationships among offshore bottlenose dolphin populations and to determine whether there are statistically significant genetic differences between them, with a particular focus on the SPSPA population. These findings will support the conservation actions needed to protect SPSPA dolphins.

## 2. Materials and Methods

### 2.1. Sample Collection

We generated a database of previously published mtDNA-CR sequences from 1485 individuals from the Atlantic Ocean (Supplementary Material Table S1) from 42 localities, with sampling location information available from previous studies [10,12,18,22,29–35]. Localities are represented in Figure 1.

Ecotype, location, sample collection methods, preservation methods, total genomic DNA extraction, amplification, and sequencing of the mtDNA control region are described in the original publications. All 1485 sequences were aligned in MEGA 11 [36] using MUSCLE [37]. Due to inconsistencies in sequence sizes, sequences were trimmed to 240 bp.

### 2.2. Genetic Diversity and Differentiation

We grouped the populations from 42 localities and tested them in different combinations based on the geographical proximity of putative populations and the population structure found in previous studies. The best combination was chosen based on the values of the analyses of molecular variance (AMOVA) in Arlequin v.3.5.2.2 [38] (Supplementary Tables S3 and S4). Significance was assessed through 10,000 permutations. The best combination was to group the populations into 15 groups (Supplementary Table S5). We removed groups with less than 10 individuals from the analysis (Supplementary Table S5), resulting in 13 groups with 1477 individuals (Table 1).

**Table 1.** Groups names and abbreviations; locations per group; number of individuals sampled (N); ecotype and previous publication. (*) Ecotype not defined by authors.

| Groups | Location | N | Ecotype | Publication |
|---|---|---|---|---|
| **Saint Peter Saint Paul Archipelago (SPSPA)** | Saint Peter Saint Paul Archipelago, Brazil | **19** | offshore | de Oliveira et al., 2019 [22] |
| **Caribbean Offshore (CAO)** | Golfo de Morrosquillo, Córdoba Province, Colombia | 3 | offshore | Caballero et al., 2012 [18] |
| | Ciénaga, Magdalena Province, Colombia | 1 | | Caballero et al., 2012 [18] |
| | Gandoca-Manzanillo, Costa Rica | 2 | | Barragan-Barrera et al., 2017 [33] |
| | Bahía de Buenavista, Cuba | 5 | | Caballero et al., 2012 [18] |
| | Between la Ceiba and Bahia de Trujillo, Honduras | 4 | | Caballero et al., 2012 [18] |
| | Puerto Rico | 20 | | Caballero et al., 2012 [18] |

**Table 1.** *Cont.*

| Groups | Location | N | Ecotype | Publication |
|---|---|---|---|---|
| | | 35 | | |
| **Gulf of Mexico Offshore (GMO)** | Gulf of Mexico, USA | 28 | offshore | Vollmer et al., 2021 [34] |
| | Holbox, Mexico | 5 | | Caballero et al., 2012 [18] |
| | Isla Mujeres, Mexico | 1 | | Caballero et al., 2012 [18] |
| | | 34 | | |
| **North Atlantic Offshore (NAO)** | The Azores, Portugal | 84 | offshore | Querouil et al., 2007 [12] |
| | Madeira, Portugal | 18 | | Querouil et al., 2007 [12] |
| | Canarias, Spain | 4 | | Fernandéz et al., 2011 [31] |
| | Pelagic Atlantic, North-East Atlantic Ocean | 101 | | Louis et al., 2014 [10] |
| | | 207 | | |
| **Caribbean Inshore (CAI)** | East Abaco, Bahamas | 29 | inshore | Parsons et al., 2006 [30] |
| | South Abaco, Bahamas | 21 | | Parsons et al., 2006 [30] |
| | White Sand Ridge, Bahamas | 5 | | Parsons et al., 2006 [30] |
| | Bahía de Buenavista, Cuba | 60 | | Caballero et al., 2012 [18] |
| | | 115 | | |
| **Gulf of Mexico Inshore (GMI)** | Gulf of Mexico, USA | 525 | inshore | Vollmer et al., 2021 [34] |
| | Celestun, Mexico | 1 | | Caballero et al., 2012 [18] |
| | Holbox, Mexico | 4 | | Caballero et al., 2012 [18] |
| | Laguna Alvarado, Mexico | 2 | | Caballero et al., 2012 [18] |
| | Laguna Terminos, Mexico | 2 | | Caballero et al., 2012 [18] |
| | Matamoros, Mexico | 4 | | Caballero et al., 2012 [18] |
| | Paraiso, Mexico | 16 | | Caballero et al., 2012 [18] |
| | Tampico, Mexico | 5 | | Caballero et al., 2012 [18] |
| | | 559 | | |
| **Namibia (NAM)** | Namíbia | **12** | * | Natoli et al., 2004 [29] |
| **Northeast of Brazil (NBR)** | Bahia, Brazil | 6 | * | de Oliveira et al., 2019 [22] |
| | Ceará, Brazil | 3 | | de Oliveira et al., 2019 [22] |
| | Pará, Brazil | 1 | | de Oliveira et al., 2019 [22] |
| | Rio Grande do Norte, Brazil | 4 | | de Oliveira et al., 2019 [22] |
| | | 14 | | |
| **Northeast Atlantic—Coastal North (NEAn)** | Shannon Estuary, Ireland | 44 | inshore | Mirimin et al., 2010 [35] |
| | Connemara–Mayo, Ireland | 12 | | Mirimin et al., 2010 [35] |
| | Cork Harbor, Ireland | 4 | | Mirimin et al., 2010 [35] |
| | Coastal North, North-East Atlantic | 76 | | Louis et al., 2014 [10] |
| | | 136 | | |
| **Northeast Atlantic—Coastal South (NEAs)** | Mauritania | 1 | inshore | Natoli et al., 2004 [29] |
| | Coastal South, North-East Atlantic | 115 | | Louis et al., 2014 [10] |
| | | 116 | | |
| **Panama (PAN)** | Bocas del Toro, Panamá | **25** | inshore | Barragan-Barrera et al., 2017 [33] |

**Table 1.** *Cont.*

| Groups | Location | N | Ecotype | Publication |
|---|---|---|---|---|
| **USA Inshore (USA)** | Charleston Harbor, USA | 35 | inshore | Richards et al., 2013 [32] |
| | Indian River, USA | 97 | | Richards et al., 2013 [32] |
| | | **132** | | |
| **South of Brazil (SBR)** | Campos and Santos Basins, Brazil | 44 | * | de Oliveira et al., 2019 [22] |
| | Northern coast of Rio Grande do Sul, Brazil | 29 | | de Oliveira et al., 2019 [22] |
| | | **73** | | |
| **USA Inshore (USA)** | Charleston Harbor, USA | 35 | inshore | Richards et al., 2013 [32] |
| | Indian River, USA | 97 | | Richards et al., 2013 [32] |
| | | **132** | | |

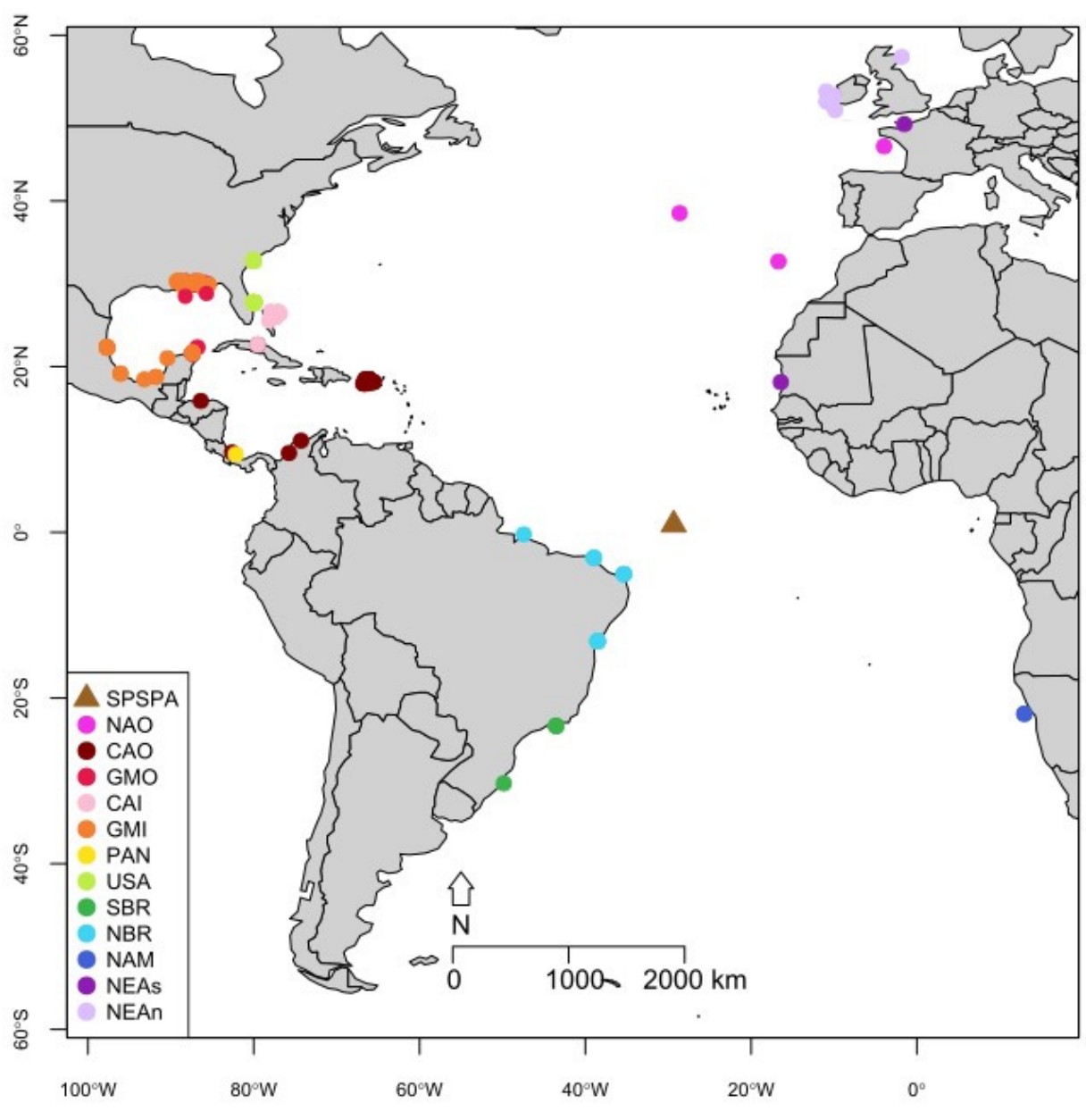

**Figure 1.** Map showing sampling locations for the bottlenose dolphin groups analyzed in this study.

Due to the differences in sampling sizes among groups, a standardized measure of genetic differentiation following Meirmans [39] was calculated in the software Genodive 3 [40] for the 13 groups and also for inshore and offshore groups separately (Table 2).

**Table 2.** AMOVA with all groups, and according to ecotypes (Inshore and Offshore groups) based on mtDNA control region.

|  | Source of Variation | *df* | % Var | *F* | *p* |
|---|---|---|---|---|---|
| All groups | Among individual | 1464 | 0.674 | 1 | 0.001 |
|  | Among population | 12 | 0.326 | 0.326 | 0.001 |
| Inshore groups | Among individual | 1105 | 0.605 | 1 | 0.001 |
|  | Among population | 6 | 0.395 | 0.395 | 0.001 |
| Offshore groups | Among individual | 291 | 0.793 | 1 | 0.001 |
|  | Among population | 3 | 0.207 | 0.207 | 0.001 |

*df*: degree of freedom; % Var: percentage of variance; *F*: fixation indices; *p*: probability of significance.

Pairwise genetic differentiation between groups was calculated in Arlequin v.3.5.2.2 [38], using fixation indices based on haplotype frequencies (FST; [41]) and genetic divergence (ΦST; [42]). Statistical significance was evaluated using the null distribution generated from 10,000 non-parametric random permutations of the data at the 0.01 significance level, and value was assessed through 10,000 permutations. A Bonferroni correction was applied for multiple comparisons using the function p.adjust, available in the stats package in software R 3.4.4 [43].

We computed haplotype and nucleotide genetic diversity for each group [44] by grouping samples based on their sampling location using DnaSP v.6 [45]. Genetic diversity indices (number of polymorphic sites, number of haplotypes, haplotype diversity, nucleotide diversity) were calculated in DnaSP v.6.

Nei's genetic divergence (Nei's dA) was estimated using the Nucleotide Divergence function of the StrataG package [46] in the software R 3.4.4 [43]. The most suitable evolution model suggested by jModeltest2 [47] was T92+G. Since this model was unavailable in the StrataG package, we used the highest-rank model (from the jModelTest output) available for subsequent analyses. Therefore, we used the Tamura-Nei gamma (TN93+G) model, with a gamma value of 0.157.

### 2.3. Demographic Equilibrium and Population Expansion

We tested the demographic equilibrium in each population by calculating Fu's Fs [48] and Tajima's D [49] statistics. Statistical significance was obtained as the proportion of simulated values smaller than or equal to the observed values ($\alpha = 0.02$ for Fu's Fs and $\alpha = 0.05$ for Tajima's D).

### 3. Results

#### 3.1. Genetic Diversity

The alignment of the mtDNA control region encompasses 240 bp, of which 43 were variable (Table 3). These polymorphic sites defined 74 distinct haplotypes (Figure 2). The summary of genetic diversity indices and neutrality tests is presented in Table 3. The NAO group has the highest haplotype (h) diversity, the NBR group has the highest nucleotide diversity, and the SPSPA population has the lowest nucleotide and haplotype diversity indices.

**Table 3.** Sample information, genetic diversity indices, and neutrality tests for populations of *Tursiops truncatus*. Number of individuals sampled (n); Number of polymorphic sites (S); Number of haplotypes (H); Haplotype diversity (h); Nucleotidic diversity (π); (SD) Standard deviation (SD). Values in **bold** indicate significance.

| Groups | n | S | H | h ± SD | π ± SD | Tajima's D | Fu's FS |
|---|---|---|---|---|---|---|---|
| Saint Peter Saint Paul Archipelago (SPSPA) | 19 | 1 | 2 | 0.105 ± 0.092 | 0.00044 ± 0.00039 | −1.16480 | −0.838 |
| Caribbean Offshore (CAO) | 35 | 11 | 7 | 0.605 ± 0.070 | 0.00735 ± 0.00174 | −1.26300 | −0.626 |
| Gulf of Mexico Offshore (GMO) | 34 | 5 | 7 | 0.838 ± 0.029 | 0.00592 ± 0.00058 | 0.40461 | −1.363 |
| North Atlantic Offshore (NAO) | 207 | 26 | 40 | 0.937 ± 0.007 | 0.02245 ± 0.00057 | 0.34314 | −13.050 |
| Caribbean Inshore (CAI) | 115 | 19 | 11 | 0.574 ± 0.048 | 0.01008 ± 0.00139 | −0.92802 | −0.456 |
| Gulf of Mexico Inshore (GMI) | 559 | 17 | 20 | 0.805 ± 0.007 | 0.00730 ± n.d. | −0.81701 | −6.136 |
| Namibia (NAM) | 12 | 6 | 3 | 0.545 ± 0.144 | 0.00989 ± 0.00283 | 0.72327 | 2.792 |
| Northeast of Brazil (NBR) | 14 | 17 | 9 | 0.912 ± 0.059 | 0.02293 ± 0.00349 | −0.02084 | −1.222 |
| Northeast Atlantic—Coastal North (NEAn) | 136 | 8 | 3 | 0.443 ± 0.040 | 0.01126 ± 0.00110 | 1.95996 | **9.058** |
| Northeast Atlantic—Coastal South (NEAs) | 116 | 7 | 3 | 0.204 ± 0.047 | 0.00159 ± 0.00061 | −1.63199 | 0.224 |
| Panama (PAN) | 25 | 0 | 1 | - | - | - | - |
| South of Brazil (SBR) | 73 | 25 | 17 | 0.888 ± 0.016 | 0.01901 ± 0.00161 | −0.59038 | −2.055 |
| USA (USA) | 132 | 4 | 5 | 0.367 ± 0.048 | 0.00189 ± 0.00028 | −0.74014 | −1.589 |
| **Total** | 1477 | 43 | 74 | 0.909 ± n.d. | 0.02230 ± n.d. | - | - |

The offshore populations from the Caribbean (CAO) and Gulf of Mexico (GMO) present slightly higher values of haplotype diversity compared to their respective inshore populations (CAI and GMI). The population from Panamá (PAN) only presented one haplotype.

Of the 74 haplotypes recovered, 59 were unique to a single location. Considering groups, the most common haplotype is Hap 2, which is found in eight groups. On the other hand, Hap 24 was the most frequent among individuals, found in 259 animals from 2 groups (Figure 2).

The median-joining network (Figure 2) indicates a division into two main clusters. The first haplotype cluster includes the inshore groups—CAI, GMI, PAN, and USA. The second haplotype cluster consists of the other inshore, offshore, and undefined ecotype groups. This second cluster has a central haplotype shared by the groups NAO, GMO, CAO, SPSPA, CAI, SBR, NBR, and NAM. This central haplotype is connected with several others (Figure 2), indicating that it is possibly an ancestral haplotype. This second haplotype group has a subgroup with exclusive haplotypes from NAO, SBR, and NBR. One haplotype is shared between NAO-NBR and NAO-CAO. The intermediate haplotypes between this subgroup and the most ancestral haplotype are from NAO, CAO, and GMO. The SPSPA population has two haplotypes; one is this ancestral haplotype, and the second is shared with NBR.

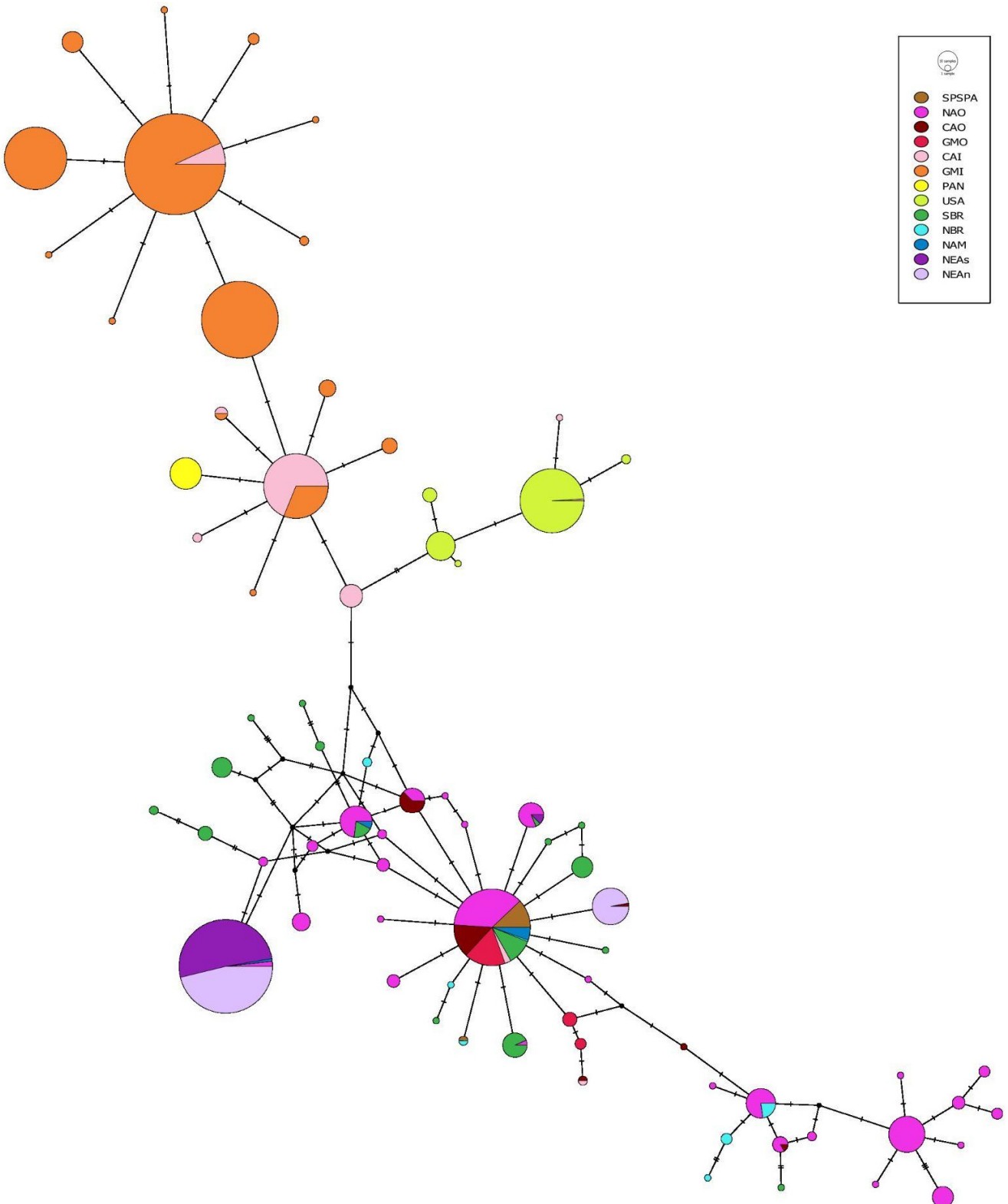

**Figure 2.** Median-joining network using 240 bp mtDNA control-region haplotypes of the bottlenose dolphin (*Tursiops truncatus*). The size of the circles is proportional to the number of samples for that haplotype. Branch lengths are proportional to the number of mutations. Colors illustrate where the haplotypes were sampled. Black dots represent inferred node haplotypes not found in the data set. Tick marks represent mutational steps. Group abbreviations are in Table 1.

### 3.2. Population Structure

The AMOVA for all groups showed a population differentiation with approximately 32% ($p < 0.01$) of the genetic variability being partitioned among the studied areas (Table 2). For inshore groups only, genetic variability was 39% ($p < 0.01$), while for offshore groups, it was 20% ($p < 0.01$). The genetic variability among individuals within the groups was 67.4% ($p < 0.01$) for all groups, 60.5% ($p < 0.01$) for inshore groups, and 79.3% ($p < 0.01$) for offshore groups.

Most of the haplotype-based indices (FST) and nucleotide-based indices (ΦST) were statistically significant ($p < 0.01$). The lowest significant ΦST value (ΦST = 0.1856, $p < 0.0001$) was obtained between the NAO and GMO offshore groups, while the highest was between SBR and GMI groups (ΦST = 1.000, $p < 0.0001$) (Figure 3). For FST indices, the lowest significant value (FST = 0.06053, $p < 0.0001$) was obtained between the SBR and NAO groups, while the highest was between PAN and NEAs groups (FST = 0.97202, $p < 0.0001$) (Figure 3).

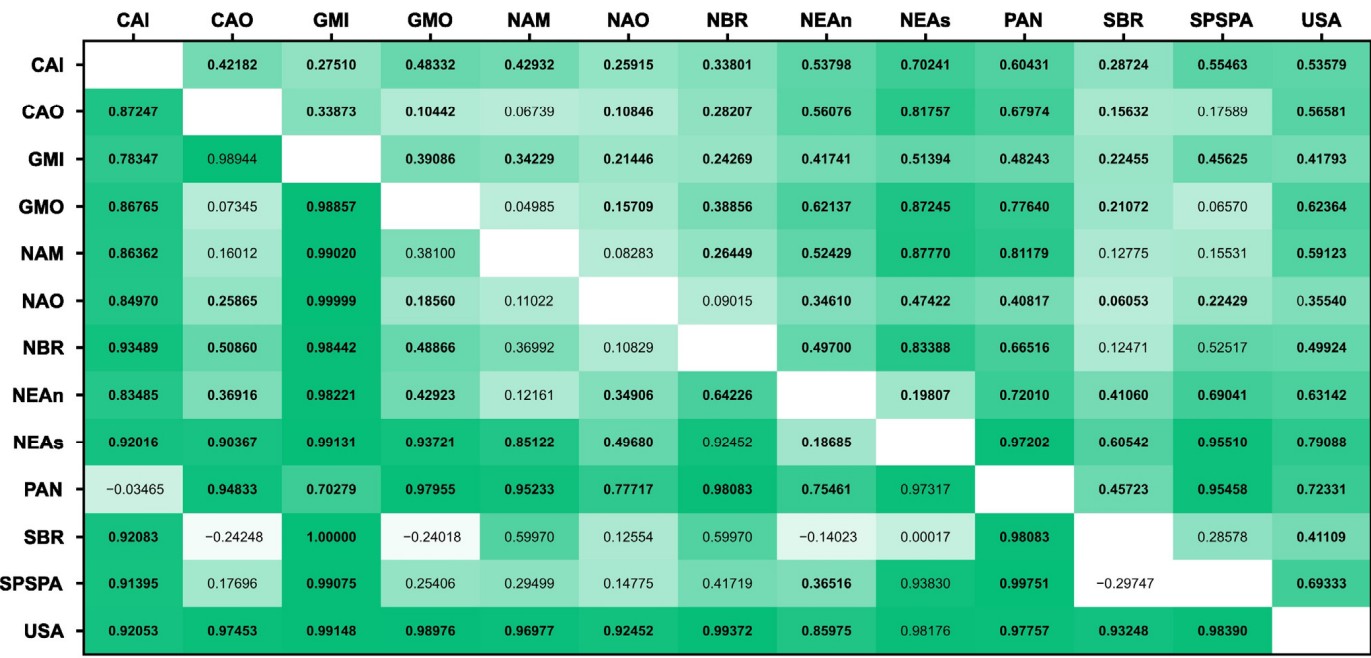

**Figure 3.** Fixation indices among populations. ΦST values are below the diagonal. FST values are above the diagonal. Values in bold indicate significance after Bonferroni correction. Cells are shaded in increasingly darker green in proportion to increasing values to aid visualization. Group abbreviations are in Table 1.

In most combinations among offshore groups, the values of ΦST were not significant, with an exception between NAO-GMO (ΦST = 0.1856, $p < 0.0001$) and NAO-CAO (ΦST = 0.25865, $p < 0.0001$). According to FST indices (Figure 3), population structure was significantly detected between the offshore groups from CAO-NAO (FST = 0.10846, $p < 0.0001$), GMO-NAO (FST = 0.15709, $p < 0.0001$), and SPSPA-NAO (FST = 0.22429, $p < 0.0001$).

Nei's dA values ranged from 0.00003 to 0.04120 (Figure 4). The lowest value obtained between groups was for CAO-NBR (Nei's dA = 0.00003). The highest value between groups was for CAI-PAN (Nei's dA = 0.04120). In offshore groups, the lowest value obtained was for NAO-GMO (Nei's dA = 0.00344), and the highest value was for CAO-GMO (Nei's dA = 0.03184).

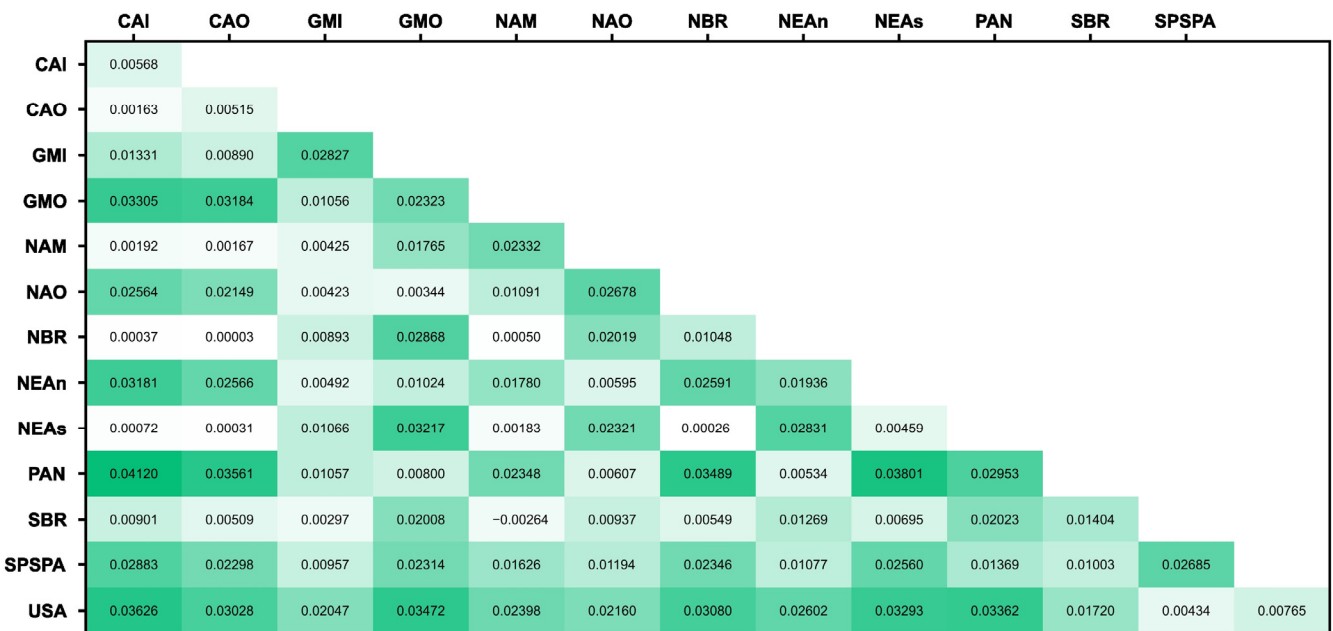

**Figure 4.** Nei's net nucleotide divergence. Divergence between populations below diagonal. Divergence within populations in diagonal. Cells are shaded in increasingly darker green in proportion to increasing values to aid visualization. Group abbreviations are in Table 1.

SPSPA's $\Phi$ST values are not significant compared with other offshore groups (CAO, GMO, and NAO) and undefined ecotype groups (NAM, SBR, and NBR); in relation to inshore groups, the values are significant and show high population differentiation. The SPSPA's FST values compared with offshore groups are only significant between the SPSPA and NAO groups ($\Phi$ST = 0.22429, $p < 0.0001$). The lowest Nei's dA value obtained in the SPSPA group is between SPSPA-GMI (Nei's dA = 0.00957) and SPSPA-SBR (Nei's dA = 0.01003), and the highest was for SPSPA-CAI (Nei's dA = 0.02883).

### 3.3. Tests of Demographic Equilibrium and Population Expansion

Fu's Fs and Tajima's D statistics were calculated for all population units defined a priori. Values of Fs and D were negative for most groups and significant only in NEAn (Fu's FS = 9.058) (Table 3). It is expected for those values to be close to zero in stable groups over time. Although most of our values were negative, these values were not significant, accepting the null hypothesis of constant group size.

### 4. Discussion

In this study, we aimed to assess the genetic differentiation of the bottlenose dolphin across its distribution in the Atlantic Ocean, using the most comprehensive mtDNA-CR dataset for this species: 1485 individuals from 42 localities. We chose mtDNA-CR as it is by far the most selected marker used in taxonomic studies of cetaceans [50]; despite being a single locus and a matrilineal marker, the popularity of mtDNA sequence data reflects the fact that this marker exhibits many attributes that make it particularly suitable for phylogeographic studies [51–53].

When aligning a large database dataset, we lose base pairs when creating the consensus sequence because many available sequences correspond to different parts of mtDNA-CR. However, even with a small consensus sequence of 240 bp, most of our results are in agreement with previous studies.

Rosel et al. [28] used mtDNA control-region sequence data from populations, sub-species, and species of cetaceans to compare several metrics of genetic differentiation. They concluded that Nei (1987), the measure of net genetic divergence (dA), and $\Phi$ST returned

the best results, exhibiting the least overlap for estimates between populations, subspecies, and species pairs of cetaceans [28].

Marko and Hart [54] concluded that the variation in isolation time and effective population size may be more important than gene flow in explaining patterns of population differentiation. Therefore, inferences based on FST can be misleading; comparisons between cetacean taxa support this conclusion [28]. A strong matrilineal-based social structure can quickly lead to fixed differences in sequence data, mainly maternally inherited ones, even in genetic exchange between populations [28].

Exploring databases has the benefit of facilitating the comparison of large datasets of populations from various locations around the world, particularly for populations that are challenging to study, such as offshore populations. This investigation, based on matrilineal genetic structure, is crucial for enhancing our understanding of the population dynamics in the Atlantic Ocean, the population structure of offshore populations, and determining which populations are most related to the population of the SPSPA. This is especially significant as the SPSPA population is situated in a central region of the Atlantic Ocean, and its conservation status is a cause for concern.

### 4.1. Insights into Offshore Populations of Bottlenose Dolphins in the Atlantic Ocean

Our results showed that offshore groups have greater genetic variability (~20%) among individuals within the group and less variation between groups compared to inshore groups (Table 2). We found higher values of population structure among coastal groups than offshore groups (Figures 3 and 4). These results indicate that offshore groups are more genetically diverse and have less population structure than inshore populations, which is consistent with previous studies [9–11,13,14].

Throughout the geographic regions included in this study, the species *T. truncate* presents considerable genetic differentiation between groups. Almost all groups defined by geographic region or habitat use (such as inshore and offshore populations) showed significant differentiation. The high level of differentiation between regional populations suggests a high potential for speciation in this genus [55]. Therefore, our results suggest an important degree of genetic structure.

In our results, the group with the highest haplotypic diversity was the NAO group (Table 3). Offshore groups from the Caribbean and Gulf of Mexico (CAO and GMO) exhibit higher haplotypic diversity than their inshore counterparts (CAI and GMI). The SPSPA population has the lowest genetic diversity in our database, which is also consistent with previous studies [15,22].

We found the division of two large groups of haplotypes in the Atlantic Ocean, one consisting of the CAI-GMI-USA-PAN haplotypes and the other with the rest of the haplotypes, both from coastal and offshore regions (Figure 2). The USA group possesses private haplotypes and shares one haplotype with the CAI group, corroborating the findings of Costa et al. [16], whose results support that the USA group is another species—*T. erebennus*. The PAN group has only one haplotype that is not shared with any other group; Barregan-Barrera et al. [33] indicate that this is a small, isolated, inshore population.

In the haplotype network, we can observe that offshore population haplotypes are spread across almost all populations. One haplotype occupies a central position, is connected with many other haplotypes, and is shared by all offshore groups (NAO, CAO, GMO, and the SPSPA), as well as the different groups of CAI, NAM, NBR, and SBR, indicating that this haplotype could be more ancestral.

Previous studies using mitogenomic data suggest that ancestral migrants of coastal bottlenose dolphins from the Western North Atlantic colonized coastal niches in the Caribbean during the late Pleistocene, around 486,000 years ago [56,57]. The genus *Tursiops* diversified in the Holocene [56], coinciding with the end of the last glacial period, approximately 27,000 to 14,000 years ago, when sea levels were low. Changes in ocean productivity and sea levels, which provided new habitats for colonization [56–59], could have influenced the distribution of coastal forms [29,56,57], supporting our results.

According to Louis et al. [14], offshore populations experienced demographic expansions approximately 150,000 to 120,000 years ago. Their analyses revealed a complex, reticulated evolutionary history of bottlenose dolphins. The offshore populations were found to be more genetically similar to the common ancestral population, while coastal populations experienced population-specific solid drift. Although none of the values were significant, offshore groups showed signs of population expansion (Table 3).

Tezanos-Pinto et al. [9] suggest that offshore and unknown ecotypes are interconnected through long-distance gene flow and/or by interchange with oceanic populations. It is not clear what evolutionary processes have led to this pattern (e.g., foraging or reproductive strategies, environmental factors, and social structure).

The genetic diversity values and haplotype distribution observed in offshore *T. truncatus* populations could reflect founding events due to the recent colonization of coastal habitats [9,18,29] on the two sides of the Atlantic basin [10]. However, according to the results of Tezanos-Pinto et al. [9], such diversity values are unlikely to persist in small, isolated populations without additional influx from other sources. The preservation of offshore populations is important because they can act as a reservoir for coastal populations and allow for recovery after dramatic events [12,29].

### 4.2. Implications for the Conservation of the SPSPA Population

Dolphins are highly mobile and capable of long-distance dispersal [29,60]. However, observational data also suggest that neither males nor females disperse far from their natal groups in the Atlantic Ocean [12,30,61]. These animals exhibit diverse site fidelity patterns; some individuals occupy large areas, while others are restricted to smaller regions. Additionally, some exhibit year-round residence patterns, while others are seasonal or transient visitors [62–66]. This variety of site fidelity is primarily attributed to the spatial and temporal predictability of available food resources [67].

Many coastal populations of bottlenose dolphins show high levels of philopatry and genetic isolation despite no apparent geographic barriers [9,30,33,68–72]. Familiarity with the natal area, particularly with foraging specialization, may also play a substantial role in reducing the dispersal tendencies of both sexes of bottlenose dolphins and in establishing population structuring on fine scales [61].

These are not exclusive characteristics of coastal populations. The SPSPA population, consisting of approximately 30 individuals, demonstrates high fidelity to the location, being considered a small resident population [17,25,73]. The lowest genetic diversity observed in this analysis may reflect a philopatry behavior.

A genetic study based solely on mtDNA-CR analysis suggested that the bottlenose dolphins from SPSPA may constitute a small and isolated population [74]. Castilhos et al. [15] analyzed microsatellite markers and mtDNA-CR (457 bp) analysis to evaluate genetic diversity, comparing the SPSPA population with inshore and offshore populations from the North Atlantic Ocean. In their analyses, they found a haplotype diversity of 0.38 and a nucleotide diversity of 0.0016 [15]. These values are lower than offshore populations from the Azores and Madeira [12], and they didn't find evidence of inbreeding in the SPSPA population [15].

To reconstruct the network, Castilhos et al. [15] used 288 bp from 57 sequences, including 17 from SPSPA individuals and 40 from inshore and offshore North Atlantic individuals. They observed that the SPSPA has two haplotypes: one haplotype shared among the SPSPA, offshore and inshore populations from the North Atlantic; and a second private haplotype from the SPSPA, as in our analyses. These results indicated that SPSPA bottlenose dolphins are part of a larger oceanic population [15].

Oliveira et al. [25] analyzed a 316 bp sequence of the mtDNA-CR from 19 individuals; their results revealed a total of two polymorphic sites defining two different haplotypes, resulting in extremely low genetic diversities (h = 0.1053 and $\pi$ = 0.00067). Oliveira et al. [25] cautioned about the small effective population size and low genetic diversity of the bot-

tlenose dolphin population from SPSPA, which are reasons for great concern regarding the protection of the SPSPA population.

Oliveira et al. [22] analyzed 316 bp alignment of 109 common bottlenose dolphin mtDNA-CR sequences from populations from the South Western Atlantic Ocean. They compared the SPSPA population with other individuals, sampled along the Brazilian coast. They observed that the diversity of haplotypes and nucleotides was the lowest in their dataset (h = 0.11 and $\pi$ = 0.007) [22]. They found two haplotypes for the SPSPA population, which were not shared with other Brazilian populations. In contrast, we observed a shared haplotype between the SPSPA and NBR, which is geographically closer to the SPSPA than the other regions analyzed.

Pratt et al. [23] used a genomic approach to compare inshore and offshore ecotype populations from Australia, New Zealand, and Brazil, including SPSPA and other offshore individuals collected in the south of Brazil. They found strong genomic differentiation between each putative lineage, suggesting that ecotypic differentiation can lead to incipient speciation. Offshore populations from across three ocean basins were found to be more genomically similar to each other than to their adjacent inshore populations [23]. In their results, Wright's inbreeding coefficient value of the SPSPA was negative, indicating that mates are, on average, less closely related than expected by chance.

Our results show that the SPSPA population has the lowest genetic diversity values, which agrees with previous studies. The most frequent haplotype in the SPSPA is shared with other offshore populations. This result supports that the SPSPA population is part of or was relatively recently colonized by migrants from a sizeable oceanic population from the North Atlantic [12,15,22].

Considering Nei's distance, the SPSPA population shows high population structure values and the lowest genetic difference compared with the GMI group and the SBR group. Oliveira et al. [22] found the lowest FST and $\Phi$ST values when comparing the SPSPA with individuals from the Campos and Santos Basins (in the SBR group). The lowest and most significant $\Phi$ST values detected were between SPSPA and NAO. Although the lowest values were detected between SPSPA-GMO and SPSPA-CAO, the values were not significant.

The SPSPA population, even though it is offshore, shares haplotypes with other populations, which suggests gene flow. However, it exhibits a relatively high population structure and the lowest values of genetic diversity among the groups analyzed, which is expected for a small population residing in an isolated archipelago.

The dolphins from the surroundings of the SPSPA have been residents of this area for 20 years, but a recent decrease in sightings of these animals has been noticed (Hoffmann et al., prep). It is possible that some unknown factor has affected their presence in the area, leading to a potential modification in their temporal and/or spatial occupation pattern. In recent years, there has been a significant increase in the Galapagos shark, *Carcharhinus galapagensis*, population around the SPSPA [75]. However, it is premature to assume that the decrease in sightings is directly related to this event. A more systematic field effort is currently underway to evaluate the extent of this population's use of the area (Hoffmann et al., in prep).

## 5. Conclusions

Offshore populations from the Ocean Atlantic have higher haplotype diversity and less variation among populations than inshore populations; these results support the idea that there is more gene flow among offshore populations, which is important because they can act as a genetic diversity reservoir for coastal populations.

Despite being an oceanic population, the SPSPA dolphins present high site fidelity and low genetic diversity. This population shares a haplotype with other offshore populations, suggesting gene flow in the present or recent past. Even so, the conservation status of this population is a major concern due to a decrease in sightings around the archipelago and its low genetic diversity compared to other populations. The SPSPA dolphins also presented concentrations of organochlorine compounds [76], with values higher than those

observed in previous studies carried out in the southeastern region of Brazil for the same species, indicating that the animals may be incorporating these organohalogens through bioaccumulation and biomagnification along food webs. Therefore, it is important to continue monitoring and studying this population. Our next steps involve conducting a genome-wide study (Alexandre et al. in prep) to gain a better understanding of the SPSPA population and its dynamics with other offshore populations.

Impacts of human-induced climate change, habitat fragmentation, and the over-exploitation of natural resources have depleted global biodiversity, particularly in the marine environment [77,78]. Offshore environments face several threats, including noise pollution from sources such as seismic surveys and active naval sonars, in addition to the loss of benthic habitat, dredging, and contamination [79].

Offshore biodiversity is frequently overlooked due to the challenges of data collection, making it difficult to assess the extent of threats to these populations. Understanding the patterns of genetic diversity is essential for gaining insights into their environmental interactions and developing more effective management strategies. This includes implementing robust biomonitoring programs to identify and mitigate ecological issues [80,81].

**Supplementary Materials:** The following supporting information can be downloaded at https://www.mdpi.com/article/10.3390/ecologies5020011/s1, Table S1: Sample Information from previous studies: accession number, locality, and ecotype; Table S2: location, acronym used in this study for location, number of individuals, and ecotype; Table S3: Information from 20 combinations tested among different locations; Table S4: AMOVA results for each tested combination (% variation, *p* value, Standard Phi-statistics); Table S5: Information on the best combination (combination 17).

**Author Contributions:** Conceptualization, B.G.A. and K.B.d.A.; methodology, B.G.A., K.B.d.A. and R.Z.; formal analysis, B.G.A. and R.Z.; investigation, B.G.A.; resources, T.R.O.d.F.; data curation, B.G.A.; writing—original draft preparation, B.G.A.; writing—review and editing, M.M.C., K.B.d.A., L.S.H., T.R.O.d.F. and R.Z.; visualization, B.G.A., R.Z., M.M.C., K.B.d.A., L.S.H. and T.R.O.d.F.; supervision, R.Z. and T.R.O.d.F.; project administration, L.S.H. and T.R.O.d.F.; funding acquisition, T.R.O.d.F. All authors have read and agreed to the published version of the manuscript.

**Funding:** This research was funded by Conselho Nacional de Desenvolvimento Científico e Tecnológico (CNPq), grant number 44325/2019-0, CNPq PhD fellowship 140281/2020-7.

**Institutional Review Board Statement:** Not applicable.

**Informed Consent Statement:** They have consented.

**Data Availability Statement:** The sequences utilized in this study are sourced exclusively from existing data available on GenBank. GenBank identifications for the specimens used are explicitly stated in the Supplementary Material (Table S1). Sample information is data published in previous studies. These sequences are accessible through the GenBank public database. All data generated or analyzed in this study are fully incorporated into this published article.

**Acknowledgments:** We would also like to thank Dra Inês Carvalho for her comments, which notably improved the manuscript. The Federal University of RioGrande do Sul (UFRGS)supported this work.We would like to thank the Brazilian Agency of the Coordination for Improvement of Higher Education Personnel (CAPES); the National Council for Scientific and Technological Development (CNPq), and the Research Support Foundation of the State of Rio Grande do Sul (FAPERGS).

**Conflicts of Interest:** The authors declare no conflicts of interest. The funders had no role in the design of the study, in the collection, analyses, or interpretation of data, in the writing of the manuscript, or in the decision to publish the results.

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
