# Peer review of "Exploring mtDNA Databases to Evaluate the Population Structure and Genetic Diversity of Tursiops truncatus in the Atlantic Ocean: Implications for the Conservation of a Small, Offshore Population"

_2673-4133, doi:10.3390/ecologies5020011_

Round 1

Reviewer 1 Report

Comments and Suggestions for Authors

Overall Evaluation:

The reviewer acknowledges the author's analysis of mitochondrial control region genetic diversity and structure in two ecotypes of bottlenose dolphins in the Atlantic. The observation that the Saint Peter Saint Paul Archipelago population has lower genetic diversity underscores the urgent need for its conservation. However, concerns are raised about potential sampling bias due to varied sources of stranding samples and the lack of adjustment for sample size effects in data analysis, raising questions about the robustness of the conclusions. The reviewer also notes that the author relies on stranding samples from previous literature, but highlights that the sources of stranding samples may differ across locations. For instance, open waters may have a more diverse sample of populations, while islands, being smaller and constrained by ocean currents, may only accept stranding samples from fixed population. This potential singularity in the source of dolphin stranding samples could lead to passive sampling bias.

Detailed Comments:

Line 25: Add abbreviations after the full name.

Line 68: Provide specific details on the extent of the decline in individual numbers.

Lines 74-79: Clarify the statement on gene flow.

Line 97: Suggest using "summarized" instead of "generated."

Line 108: The mitochondrial control region gene fragment length of 240 bp may be too short.

Lines 127-131: Recommend trying alternative software for analysis, such as MEGA.

Line 135: Italicize Orcinus orca.

Lines 140 and 151: Add a space between "140 bp."

Figure 2: Use different color schemes for offshore and nearshore populations, and adjust the legend size for clarity.

Figure 4: Group data of the same ecological type together for ease of comparison.

Table 1: Address abbreviation issues, remove the redundant Haplotypes column, and consider differentiating abbreviations with I or O to denote ecological types.

Tables 3 and 4: Consider using color-coded charts for better visual interpretation.

Author Response

Response to Reviewer 1:      

We would like to thank you for the time and effort of the reviewers, the suggestions were crucial to improve the manuscript. That said, we made some modifications to the research design, methodology, results, and discussion taking your considerations into account.  The alterations in the text are in blue. Since we made changes in the research design and methods, the figures and tables are new, although the general content is very similar to the previous version.  The reviewers addressed some comments and suggestions, which we answered point-by-point in the document below, in blue.

Comments:

Comments 1: Overall Evaluation:

The reviewer acknowledges the author's analysis of mitochondrial control region genetic diversity and structure in two ecotypes of bottlenose dolphins in the Atlantic. The observation that the Saint Peter Saint Paul Archipelago population has lower genetic diversity underscores the urgent need for its conservation. However, concerns are raised about potential sampling bias due to varied sources of stranding samples and the lack of adjustment for sample size effects in data analysis, raising questions about the robustness of the conclusions. The reviewer also notes that the author relies on stranding samples from previous literature, but highlights that the sources of stranding samples may differ across locations. For instance, open waters may have a more diverse sample of populations, while islands, being smaller and constrained by ocean currents, may only accept stranding samples from fixed populations. This potential singularity in the source of dolphin stranding samples could lead to passive sampling bias.

Thank you for pointing this out. We agree with this comment. We understand the concerns and limitations of this analysis. We took into account the reviewer's suggestions and we revised the data to affirm the ecotype only when the author of the previous study was able to identify it (morphologically or genetically) and not based on locality. Also, we redid the analyses adjusting the sample sizes following the methodology used by Amaral et al. 2021.  We know the difficulties in working with animals such as cetaceans, due to the difficulty in obtaining samples, and often, the information comes from stranding samples. Taking all these notes into consideration, and also knowing that the SPSPA population has characteristics somewhat different from other offshore populations, we restructured this study, from methodology to discussion, avoiding possible hasty conclusions.

Comments 2: Line 25: Add abbreviations after the full name.

Corrected, the abbreviation is found on line 27.

 Comments 3: Line 68: Provide specific details on the extent of the decline in individual numbers.

We rewrote this paragraph from the introduction to make this information clearer to the reader (line 100-103). In our last expeditions to SPSPA, we no longer found the dolphin population. However, we want to avoid any erroneous and hasty indication, before we finish collecting data. At this moment our research group is working on this study to evaluate the presence and absence of animals around the archipelago.

Comments 4: Lines 74-79: Clarify the statement on gene flow.

 We rewrote the sentence in line 74-77 to clarify what gene flow would be, the sharing of haplotypes indicates the exchange of genetic material between populations, in the present or in the near past.

 Comments 5: Line 97: Suggest using "summarized" instead of "generated."

This paragraph was modified.

Comments 6: Line 108: The mitochondrial control region gene fragment length of 240 bp may be too short.

 We understand that our limitation in this study is the size of our sequence, which was a consequence of having aligned this large number of samples. Previous studies used a larger sequence with a smaller number of individuals from a few populations. The novelty of this work is to compare the largest data set for this species.

Comments 7: Lines 127-131: Recommend trying alternative software for analysis, such as MEGA.

We appreciate the suggestion, but we decided to keep using the stats package in software R 3.4.4 to calculate Bonferroni correction since it is a widely used option. 

Comments 8: Lines 140 and 151: Add a space between "140 bp."

 Corrected.

Comments 9: Figure 2: Use different color schemes for offshore and nearshore populations, and adjust the legend size for clarity.

 We tried to color-code the offshore and inshore populations, using different shades of the same color for each of them.  This resulted in a figure with many similar colors, and it was hard to differentiate the individual populations. Therefore, we chose to keep a similar color scheme as in the first version of the manuscript, but we adjusted the legend size and used easily identifiable colors for the populations.  

Comments 10: Figure 4: Group data of the same ecological type together for ease of comparison.

 Corrected.

Comments 11: Table 1: Address abbreviation issues, remove the redundant Haplotypes column, and consider differentiating abbreviations with I or O to denote ecological types.

 We modified this table, removed the haplotypes columns,added a new column with the original citation of the population,  and specified the ecotypes. 

Comments 12: Tables 3 and 4: Consider using color-coded charts for better visual interpretation.

We tried, but we preferred to keep the same style as in the first version of the manuscript.

Reviewer 2 Report

Comments and Suggestions for Authors

-        The manuscript cannot be accepted as a regular article. It is a review of published data.

The manuscript in some aspects replicates the results of previous articles.

 -        The title does not mirror the manuscript. The Authors have downloaded and analyzed sequences of both inshore and offshore T.truncatus, and do not widely discuss conservation topics.

 -        The Saint Peter Saint Paul Archipelago (SPSPA) should be highlighted in the map of figure1 and clearly associated with the SAO group.

 -        The authors write that the study focuses on SPSPA populations and that their findings will define the conservation actions needed to protect SPSPA dolphins, however the sequences from SPSPA are only 19, a limited portion compared to the total sequences (1485), and scarce information about prior data have been given.

 -        Some affirmations of the authors are contrasting.

The offshore population from South Atlantic Offshore (SAO) represents the population from SPSPA. Results indicate an absence of genetic structure between the SAO offshore population and the others.

Line 341: “Our results suggest some level of genetic and social isolation,”; while, in line 355 “Despite being an offshore population, the SPSPA population presents high site fidelity and low genetic diversity. This population has low haplotype diversity and shares a haplotype with offshore populations, indicating that there might have been gene flow.”

 -        Line 245: “Previous studies have shown that offshore populations have greater genetic diversity than inshore 246 populations [12, 9, 26, 10, 13, 14, 46]. However, our results showed that the genetic diversity found in some populations, such as the SPSPA population, was lower than in previous publications [20]. This is probably due to the shorter length (240bp) of our consensus sequence compared to the original publications. We created a large dataset, sometimes the sequences are from different parts of the control region.”

Why a shorter sequence can support your results?

It is not clear the following sentence: “We created a large dataset, sometimes the sequences are from different parts of the control region.”

-        There are so many papers reporting genetic data on T.truncatus  (for instance, the below papers not cited) that the authors could write a more extensive article.

de Oliveira, L. R., Ott, P. H., Moreno, I. B., Tavares, M., Siciliano, S., & Bonatto, S. L. (2017). Effective population size of an offshore population of bottlenose dolphins, Tursiops truncatus, from the São Pedro and São Paulo Archipelago, Brazil. Latin American Journal of Aquatic Mammals, 11(1-2), 162-169. https://doi.org/10.5597/00225

Fruet, P. F., Secchi, E. R., Daura-Jorge, F., Vermeulen, E., Flores, P. A., Simoes-Lopes, P. C., ... & Möller, L. M. (2014). Remarkably low genetic diversity and strong population structure in common bottlenose dolphins (Tursiops truncatus) from coastal waters of the Southwestern Atlantic Ocean. Conservation Genetics, 15, 879-895.

 -        They should also consider accurately the results and discussions of previous articles.

Line 288: “Previous studies compared coastal and offshore western South Atlantic populations and found a high population structure between the two ecotypes and high genetic diversity in the offshore population [50, 51], unlike what we found in the resident offshore South Atlantic population of SPSPA.” In this sentence, the authors cite [51] Costa et al 2022, but this paper does not report data from South Atlantic but rather from the North Atlantic.

Further, Costa et al 2022, and other papers, discuss the existence of a different species, Tursiops erebennus, a lineage that is not cited in the present manuscript.

On the whole, the manuscript should be deeply revised and resubmitted.

Comments on the Quality of English Language

 Moderate editing of English language required

Author Response

Response to Reviewer 2:      

We would like to thank you for the time and effort of the reviewers, the suggestions were crucial to improve the manuscript. That said, we made some modifications to the research design, methodology, results, and discussion taking your considerations into account.  The alterations in the text are in blue. Since we made changes in the research design and methods, the figures and tables are new, although the general content is very similar to the previous version.  The reviewers addressed some comments and suggestions, which we answered point-by-point in the document below, in blue.

Comments:

Comments 1: The manuscript cannot be accepted as a regular article. It is a review of published data.

We understand that using a database without adding new sequences can be seen as a review- However, this study groups a new data set and performs analyses that have not been done before- For this, we believe that this is why this is an article and not a review. Unfortunately, during the pandemic years, many PhD students, as is the case of the first author, were unable to work in the field or use the laboratory; performing new analyses with databases was the solution found. Even though it is a small study, and the results only corroborate previous studies, it is relevant to explore this large set of data, which had not been done before.

Comments 2: The manuscript in some aspects replicates the results of previous articles.

Many results found in the analyses are in agreement with previous studies, however comparing new populations that had not been compared before is a relevant result.

Comments 3: The title does not mirror the manuscript. The Authors have downloaded and analyzed sequences of both inshore and offshore T.truncatus, and do not widely discuss conservation topics.

Title was rewritten to "Exploring the mtDNA Databases to Evaluate Population Structure and Genetic Diversity of Tursiops truncatus in the Atlantic Ocean: Implications for the Conservation of a Small Offshore Population".

Comments 4: The Saint Peter Saint Paul Archipelago (SPSPA) should be highlighted in the map of figure1 and clearly associated with the SAO group.

Map has been redone.

Comments 5: The authors write that the study focuses on SPSPA populations and that their findings will define the conservation actions needed to protect SPSPA dolphins, however the sequences from SPSPA are only 19, a limited portion compared to the total sequences (1485), and scarce information about prior data have been given.

The population of São Pedro São Paulo Archipelago is approximately 30 individuals. Therefore, 19 individuals is a good representation of the population. Also, collecting biological material in this area involves a huge logistic effort. We reformulated the analyses to correct the different sample sizes, following the methodology of the study published by a co-author (do Amaral et al., 2021), which also used mtDNA databases to compare populations with different sizes from different locations in the Atlantic Ocean.

Comments 6: Some affirmations of the authors are contrasting.

The offshore population from South Atlantic Offshore (SAO) represents the population from SPSPA. Results indicate an absence of genetic structure between the SAO offshore population and the others.

Line 341: “Our results suggest some level of genetic and social isolation,”; while, in line 355 “Despite being an offshore population, the SPSPA population presents high site fidelity and low genetic diversity. This population has low haplotype diversity and shares a haplotype with offshore populations, indicating that there might have been gene flow.”

Line 245: “Previous studies have shown that offshore populations have greater genetic diversity than inshore 246 populations [12, 9, 26, 10, 13, 14, 46]. However, our results showed that the genetic diversity found in some populations, such as the SPSPA population, was lower than in previous publications [20]. This is probably due to the shorter length (240bp) of our consensus sequence compared to the original publications. We created a large dataset, sometimes the sequences are from different parts of the control region.”

Taking into consideration the reviewers' suggestions and recommendations, we deeply revised the methodology and rewrote the discussion to clarify these aspects. Our objective was to group the largest data set to better understand the relationships between offshore populations compared to inshore populations in the Atlantic Ocean, and try to understand, by grouping a greater number of sequences, which populations are most related to the SPSPA population. The results regarding SPSPA remain subject of investigation, because despite being an oceanic population that shares haplotypes with other offshore populations, it is a small population with low genetic diversity. 

Comments 7: Why a shorter sequence can support your results?

We understand that our limitation in this study is the size of our sequence, which was a consequence of having aligned this large number of samples. Previous studies used a larger sequence with a smaller number of individuals from a few populations (compared to our number). The novelty of this work is to compare the largest data set for this species. However, even with a small consensus sequence of 240 bp, most of our results are in agreement with previous studies.

Comments 8: It is not clear the following sentence: “We created a large dataset, sometimes the sequences are from different parts of the control region.”

The sentence was rewritten (line 267-269).

Comments 9: There are so many papers reporting genetic data on T.truncatus  (for instance, the below papers not cited) that the authors could write a more extensive article.

de Oliveira, L. R., Ott, P. H., Moreno, I. B., Tavares, M., Siciliano, S., & Bonatto, S. L. (2017). Effective population size of an offshore population of bottlenose dolphins, Tursiops truncatus, from the São Pedro and São Paulo Archipelago, Brazil. Latin American Journal of Aquatic Mammals, 11(1-2), 162-169. https://doi.org/10.5597/00225

Fruet, P. F., Secchi, E. R., Daura-Jorge, F., Vermeulen, E., Flores, P. A., Simoes-Lopes, P. C., ... & Möller, L. M. (2014). Remarkably low genetic diversity and strong population structure in common bottlenose dolphins (Tursiops truncatus) from coastal waters of the Southwestern Atlantic Ocean. Conservation Genetics, 15, 879-895.

We include the paper by Oliveira et al. (2016) in our discussion (line 385-390), thanks for the suggestion. The dataset from paper Fruet et al. (2014) cannot be included in this analysis due to the lack of necessary information, such as number of individuals per haplotype, however, we cited the paper by Fruet et al. (2017) (line 84). 

 Comments 10: They should also consider accurately the results and discussions of previous articles.

Thank you for the suggestion, we seek to improve our discussion of the results of previous studies.

Comments 11: Line 288: “Previous studies compared coastal and offshore western South Atlantic populations and found a high population structure between the two ecotypes and high genetic diversity in the offshore population [50, 51], unlike what we found in the resident offshore South Atlantic population of SPSPA.” In this sentence, the authors cite [51] Costa et al 2022, but this paper does not report data from South Atlantic but rather from the North Atlantic.

There was an error typing the year, we have corrected the error.

 Comments 12: Further, Costa et al 2022, and other papers, discuss the existence of a different species, Tursiops erebennus, a lineage that is not cited in the present manuscript.

We include in our introduction and discussion this work and its results.

Response to Comments on the Quality of English Language

The paper was reviewed by one of our co-authors who is fluent in English.

Round 2

Reviewer 2 Report

Comments and Suggestions for Authors

The Authors followed the suggestions and improved the manuscript.

The manuscript can be accepted in its present form according to the final evaluation of the Editor.